# *GJB2* Is a Major Cause of Non-Syndromic Hearing Impairment in Senegal

**DOI:** 10.3390/biology11050795

**Published:** 2022-05-23

**Authors:** Yacouba Dia, Samuel Mawuli Adadey, Jean Pascal Demba Diop, Elvis Twumasi Aboagye, Seydi Abdoul Ba, Carmen De Kock, Cheikh Ahmed Tidjane Ly, Oluwafemi Gabriel Oluwale, Andrea Regina Gnilane Sène, Pierre Diaga Sarr, Bay Karim Diallo, Rokhaya Ndiaye Diallo, Ambroise Wonkam

**Affiliations:** 1Division of Human Genetics, Faculty of Medicine, Pharmacy and Odontology, University Cheikh Anta Diop (UCAD), Dakar 10700, Senegal; yacouba.dia@ucad.edu.sn (Y.D.); jeanpascaldemba.diop@ucad.edu.sn (J.P.D.D.); banourrou@gmail.com (S.A.B.); cheikh9611@gmail.com (C.A.T.L.); andreareginagnilane.sene@ucad.edu.sn (A.R.G.S.); lordpeter.mcsarr@gmail.com (P.D.S.), rokhaya.ndiaye@ucad.edu.sn (R.N.D.); 2Division of Human Genetics, Faculty of Health Sciences, University of Cape Town, Cape Town 7925, South Africa; smadadey@st.ug.edu.gh (S.M.A.); etaoagye@st.ug.edu.gh (E.T.A.); carmen.dekock@uct.ac.za (C.D.K.); femi.oluwole@gmail.com (O.G.O.); 3West African Centre for Cell Biology of Infectious Pathogens (WACCBIP), University of Ghana, Legon, Accra P.O. Box LG 54, Ghana; 4Department of Oto-Rhino-Laryngology, Albert Royer Children’s Hospital, Dakar 10700, Senegal; bayiallo@yahoo.fr; 5McKusick-Nathans Institute and Department of Genetic Medicine, Johns-Hopskins University School of Medicine, Baltimore, MD 21205, USA

**Keywords:** hearing impairment, *GJB2*, *GJB2*: c.94C>T: p.(Arg32Cys), Senegal, Africa

## Abstract

**Simple Summary:**

The prevalence of *GJB2*-related (MIM: 121011) congenital non-syndromic hearing impairment (NSHI) accounts for close to 50% in populations of Asian and European ancestry. However, in sub-Saharan Africa, except for Ghana, previous data showed that the prevalence of *GJB2*-associated NSHI is close to zero. To investigate the contribution of *GJB2* mutations in autosomal recessive NSHI in Senegal, we screened 129 affected and 143 unaffected individuals from 44 multiplex families, 9 sporadic cases, and 148 hearing controls with no personal or family history of hearing impairment, by targeted gene sequencing. We identified three pathogenic *GJB2* variants in 34% (*n* = 15/44) of multiplex families, of which 80% (*n* = 12/15) were consanguineous. The most common variant, *GJB2*: c.94C>T: p.(Arg32Cys), accounted for 27.3% (*n* = 12/44) of familial cases. We also identified the previously reported “Ghanaian” founder variant, *GJB2*: c.427C>T: p.(Arg143Trp), in four multiplex Senegalese families. Relatively high allele frequencies of c.94C>T. and c.427C>T variants were observed among the screened hearing controls: 1% (*n* = 2/148 ∗ 2), and 2% (*n* = 4/148 ∗ 2), respectively. No *GJB6*-D13S18 deletion was identified in any of the hearing-impaired participants. The data suggest that *GJB2*: c.94C>T: p.(Arg32Cys) should be routinely tested in NSHI in Senegal.

**Abstract:**

This study aimed to investigate *GJB2* (MIM: 121011) and *GJB6* (MIM: 604418) variants associated with familial non-syndromic hearing impairment (HI) in Senegal. We investigated a total of 129 affected and 143 unaffected individuals from 44 multiplex families by segregating autosomal recessive non-syndromic HI, 9 sporadic HI cases of putative genetic origin, and 148 control individuals without personal or family history of HI. The DNA samples were screened for *GJB2* coding-region variants and *GJB6*-D3S1830 deletions. The mean age at the medical diagnosis of the affected individuals was 2.93 ± 2.53 years [range: 1–15 years]. Consanguinity was present in 40 out of 53 families (75.47%). Variants in *GJB2* explained HI in 34.1% (*n* = 15/44) of multiplex families. A bi-allelic pathogenic variant, *GJB2*: c.94C>T: p.(Arg32Cys) accounted for 25% (*n* = 11/44 families) of familial cases, of which 80% (*n* = 12/15) were consanguineous. Interestingly, the previously reported “Ghanaian” founder variant, *GJB2*: c.427C>T: p.(Arg143Trp), accounted for 4.5% (*n* = 2/44 families) of the families investigated. Among the normal controls, the allele frequency of *GJB2*: c.94C>T and *GJB2*: c.427C>T was estimated at 1% (2/148 ∗ 2) and 2% (4/148 ∗ 2), respectively. No *GJB6*-D3S1830 deletion was identified in any of the HI patients. This is the first report of a genetic investigation of HI in Senegal, and suggests that *GJB2*: c.94C>T: p.(Arg32Cys) and *GJB2*: c.427C>T: p.(Arg143Trp) should be tested in clinical practice for congenital HI in Senegal.

## 1. Introduction

Congenital hearing impairment (HI) remains the most disabling condition with the highest rate of age-standardized disability life years [1,2]. Late diagnosis (after 2 years) results in significant sequelae with consequences for language acquisition and cognitive development [3]. The incidence of congenital HI has been estimated at 1 in 1000 live births in developed countries, but a six times higher incidence was observed in sub-Saharan African (SSA) countries [4]. Genetic factors account for 50% of congenital HI cases [5], of which 70% are non-syndromic [6]. Non-syndromic hearing impairment (NSHI) is genetically highly heterogeneous. To date, approximately 170 loci have been mapped and 124 genes have been identified [7]. The DFNB1 locus for autosomal recessive non-syndromic hearing impairment (ARNSHI) was mapped to the 13q11-q12 region [8]. This locus contains the *GJB2* and *GJB6* genes, which encode connexin 26 (Cx26) and connexin 30 (Cx30), respectively. Pathogenic *GJB2* variants are the most common genetic etiology of ARNSHI [9]. The contribution of the *GJB2* variants to ARNSHI varies from 0 to 50% in diverse populations [9]. In European and Asian populations, *GJB2* variants are the major contributors to ARNSHI [10,11]. However, except for Ghana where the *GJB2*: c.427C>T: p.(Arg143Trp) founder variant is highly prevalent [12], the prevalence of *GJB2*-related ARNSHI is close to zero in several SSA populations (Cameroon, South Africa, Nigeria, Sudan and Kenya) [13,14,15,16].

In European populations, up to 50% of individuals with ARNSHI have a pathogenic variant in the *GJB2*-coding region (exon2) at a heterozygous state [17]. It was suggested that there could be other pathogenic variants in the DFNB1 locus but outside the *GJB2* gene. This hypothesis was supported by the finding of a large genomic deletion in the DFNB1 locus outside *GJB2,* which removes the neighboring *GJB6* gene [18], which encodes Cx30, another subunit of the gap-junction channels of the auditory hair cells of the cochlea [19]. Several deletions have been reported [17,20,21]. The largest genomic deletion (342 kb), named del(*GJB6*-D13S1830), was found in up to 9.7% of affected individuals, either in a homozygous state or a heterozygous state with a GJB2 variant in trans, and constitutes the second most common genetic etiology of ARNSHI. This deletion disrupts *GJB2* expression at the transcriptional level by removing putative cis-regulatory elements upstream of *GJB6* [22].

The genetic etiology of ARNSHI in Senegal has not been investigated to date. In the present study, we examined the contribution of *GJB2* variants and del(*GJB6*-D13S1830) to ARNSHI in Senegal.

## 2. Materials and Methods

### 2.1. Ethical Approvals

The study was performed in accordance with the Declaration of Helsinki regarding medical research on humans. Ethical approval was obtained from the Research Ethics Committee of Cheikh Anta Diop University (CER/UCAD/AD/MSN/034/2020), Dakar, Senegal, and the University of Cape Town, Faculty of Health Sciences’ Human Research Ethics Committee (HREC 104/2018). Written informed consent was obtained from all the adult participants and from the parents or guardians of the minors.

### 2.2. Study Population

Hearing-impaired patients were recruited from eleven out of the fourteen administrative regions of Senegal, from children’s hospitals, schools for the deaf, as well as from the community by following the procedures previously described in Cameroon and Mali [16,23]. A total of 129 affected and 143 unaffected individuals from 44 multiplex families, segregating ARNSHI, and 9 simplex families with suspected genetic origin of HI were recruited for *GJB2* and *GJB6* genetic analyses. Pedigrees were drawn for each family through at least three generations.

We performed an otoscopic examination for all the study participants and the cerumen plug was removed before audiological evaluation. The hearing assessments were based on the international standard ISO 8253-1 [24]. The pure tone audiometry (PTA) was performed to evaluate air conduction (250 Hz to 8000 Hz) and bone conduction (250 Hz to 4000 Hz) with a mobile audiometer (KUDUWAVE TM N°0901-04011, Cape Town, South Africa). The hearing threshold was calculated as the average hearing level at 0.5, 1.0, 2.0, and 4.0 kHz. The WHO Global Burden Disease Hearing Loss Expert Group guidelines [25] were used to categorize patients according to the degree of HI. Normal hearing was defined as hearing thresholds up to 25 dB. For children who were too young for a PTA testing, auditory brainstem response (ABR) was performed when applicable.

We also recruited 148 unrelated apparently healthy individuals, who were ethno-linguistically matched, during a blood donation, from four blood banks in four administrative regions of Senegal. A questionnaire was administered to each participant for the exclusion of any personal or familial history of HI.

### 2.3. Mutation Screening of GJB2 and GJB6

Genomic DNA was extracted from peripheral blood samples, following the manufacturer’s instructions (Puregene Blood Kit^®^, (Qiagen, Alameda, CA, USA)), at the Division of Human Genetics, Faculty of Medicine, Pharmacy and Odontology of Cheikh Anta Diop University, Dakar, Senegal.

Previously reported primers for *GJB2* exon 2 [26] were evaluated with BLAST^®^ software to assess primer specificity. The coding exon of the *GJB2* gene (exon 2) was amplified, followed by Sanger sequencing in an ABI 3130XL Genetic Analyzer^®^ (Applied Biosystems, Waltham, MA, USA), at the Division of Human Genetics, University of Cape Town, South Africa. The housekeeping strategy was to sequence the coding region of *GJB2* using previously described primers for all recruited probands and affected kindreds. When a pathogenic variant was identified, we sequenced all the other family members to make sure that the identified pathogenic variant was segregated with the HI phenotype.

Subsequently, the detection of del (*GJB6*-D13S1830) was examined using the previously reported primers *GJB6*-1R (forward) and BKR-1 (reverse) [18] to amplify a 460 bp fragment corresponding to the sum of 244 bp and 216 bp, flanking the deletion, as well as a second reverse primer, *GJB6*-2R (5′-TCATCGGGGGTGTCAACAAACA-3′) that is located in the deleted segment, in order to positively detect a 681 bp fragment corresponding to the wild-type product [17].

### 2.4. Bioinformatic and Statistical Analyses

The AB1 files retrieved from the ABI 3130XL Genetic Analyzer^®^ were manually reviewed using FinchTV v1.4.0, and aligned in UGENE v34.0 [27], to a *GJB2* reference sequence [28] (NM_004004.6.; retrieved from NCBI browser). Detected variations were described using Human Genome Variation Society (HGVS) nomenclature [29], and classified using American Society of Medical Genetics’ (ACMG) guidelines [30,31]. The association between allele frequency in affected individuals and controls was assessed using the Chi-square test when applicable, or Fisher’s exact test. A p-value less than 0.05 was considered as significant. Statistical analyses were performed using R software v 4.0.5 (R Core Team, 2020. Vienna, Austria).

## 3. Results

### 3.1. Socio-Demographic Data

A total of 129 HI participants belonging to 44 unrelated multiplex families, segregating ARNSHI, and 9 simplex families with a suspected genetic origin of HI were recruited. The average number of participants from whom whole-blood samples were obtained per family was 6 and 3 for multiplex and simplex families, respectively. Consanguinity was present in 40 out of 53 families (75.47%).

The mean age of hearing-impaired participants was 14.80 ± 9.80 years [1–16 years], with a sex ratio of 0.98 (64 males and 65 females). The mean age at medical diagnosis was 2.93 ± 2.53 years [1–15 years].

### 3.2. Audiological Patterns

Grouping the 124 patients according to the degree of HI showed that the majority (102/124) had a profound HI. Two patients were too young for PTA testing (<2 years), and three patients were not available during the audiological assessment. The age at medical diagnosis was inversely correlated to the degree of HI. Profound HI was associated with an early diagnosis compared to severe and moderate HI (Table 1). The audiometric curve-pattern analysis showed a flat curve in 93 out of 124 patients (75%) and sloping in 31 patients (25%).

### 3.3. Molecular Analysis of GJB2 and GJB6

We screened 129 participants from 44 multiplex families and 9 individuals from simplex families, living with congenital sensorineural HI for variants in the coding region of the *GJB2* gene and *GJB6*-D3S1830 deletion. We did not observe any *GJB6*-D3S1830 deletion in any HI patients (Appendix A).

Three variants in *GJB2* were identified and classified as pathogenic based on the American College of Medical Genetics (ACMG) guidelines (Appendix A). Thirty-four percent (34%, *n* = 15/44) of multiplex families were positive for a *GJB2* pathogenic variant either in a homozygous state or in a compound heterozygous state. The consanguinity rate among *GJB2*-positive families was estimated at 80% (*n* = 12/15). The most common variant, *GJB2*: c.94C>T: p.(Arg32Cys), was in a homozygous state in patients from 11 multiplex families (Table 2).

*GJB2*: c.427C>T: p.(Arg143Trp) and *GJB2*: c.9C>T: p.(Arg32Cys) segregated with HI either in a homozygous state or a compound heterozygous state. Figure 1 shows the segregation of *GJB2*: c.427C>T: p.(Arg143Trp) and *GJB2*: c.9C>T: p.(Arg32Cys) in a homozygous state in two HI multiplex families.

In a particularly large multiplex consanguineous family, some patients were compound heterozygous for *GJB2* variants ([c.427C>T: p.(Arg143Trp)]; [c.94C>T: p.(Arg32Cys)], e.g., proband V.6 and her sister V.7 (Figure 2A), while in other branches of the family, different biallelic homozygous variants were found, e.g., cousin (V.1) from the father side and V.11 from the mother side (Figure 2A). The proband as well as her sister presented profound HI (Figure 2B). The proband’s affected uncle (IV.7), aunt (IV.8) and cousin (V.12) were heterozygous for the *GJB2* variant (C/T), and his cousin (V.11) was homozygous for the reference allele (Figure 2A), suggesting the implication of another gene in this particular family that can benefit from Whole-Exome-Sequencing (WES) analysis.

No pathogenic variant of *GJB2* was identified in nine individuals from simplex families with HI.

The three *GJB2* pathogenic variants identified in HI patients were also observed in a heterozygote state in the control population. The *GJB2*: c.427C>T: p.(Arg143Trp) variant is the most frequent (2%; *n* = 4/148 ∗ 2), followed by the c.94C>T: p.(Arg32Cys) variant (1%; *n* = 2/148 ∗ 2) (Table 3).

### 3.4. Phenotype-Genotype Correlation

It appears that *GJB2*: c.427C>T: p.(Arg143Trp) in the homozygous or compound heterozygous state was associated with profound HI. Only patients with *GJB2*: c.94C>T: p.(Arg32Cys) in a homozygous state showed different degrees of HI (Table 4).

## 4. Discussion

To the best of our knowledge, this is the first genetic study of ARNSHI in Senegal, which revealed a surprisingly high proportion (34%, *n* = 15/44) of pathogenic variants in *GJB2* associated with non-syndromic congenital HI. Until recently, Ghana was the exception in SSA, where *GJB2* was a major cause of HI. In light of our findings, Senegal is the second SSA country where *GJB2* variants significantly contribute to ARNSHI.

The high implication of the *GJB2* variants in ARNSHI in Senegal could be supported by the relatively high allele frequency of c.427C>T: p.(Arg143Trp) and c.94C>T: p.(Arg32Cys) in the hearing controls. The carrier frequency of c.427C>T: p.(Arg143Trp) in a control population from Ghana was estimated at 1.4% [12], which is almost half of what we have reported in Senegal (2.7%). This might be due to the control participants being recruited from geographic regions where only 1/3 of affected participants were recruited, therefore not representative of the general population of Senegal. Indeed, the recruitment of cases was based on families segregating HI in at least two affected individuals, and families were recruited nationwide from schools for the deaf, and within the communities, following similar successful methods we previously implemented in both Cameroon and Mali [16,23]. Therefore, we do not expect any significant bias in the sampling of cases. Nevertheless, the recruitment of apparently healthy controls from blood donors did not match the geographical area where families were recruited. Therefore, the ethno-linguistic and geographical origin of controls were likely not representative of the general Senegalese population, and probably biased the carrier-frequency estimates for *GJB2*-427C>T: p.(Arg143Trp), and c.94C>T: p.(Arg32Cys). This limitation should be alleviated in future studies.

The mean age at medical diagnosis of HI participants was estimated at 2.80 ± 2.53 years. A similar result has been reported in Cameroon by Wonkam et al., with a mean age at medical diagnosis of 3.2 years [16], while a higher mean age of 6 years was reported in Ghana [12]. In Mali, the median age at diagnosis was 12 years [23]. In contrast to SSA, a mean age at diagnosis of less than 6 months has been reported in the United States [32], and between 12 and 36 months in France depending on the degree of HI [33]. This disparity could be explained by limited universal newborn hearing screening (UNHS) in most SSA countries, and none in others, e.g., in Senegal [34]. We also observed an inverse correlation between age at diagnosis and the degree of HI, as previously reported elsewhere [33].

In this study we identified a common variant, *GJB2*: c.94C>T: p.(Arg32Cys), in 12/44 of multiplex families. Families positive for this variant were recruited across the country, from the western, northern and central geographic regions of Senegal. Ely CMM et al. reported this variant in two consanguineous families in Mauritania, a northern neighboring country of Senegal [35]. *GJB2*: c.94C>T: p.(Arg32Cys) has also been reported in hearing-impaired individuals in China [36], Japan [37], and South Korea [38]. Owing to the high positivity rate, it might be worth developing an affordable diagnostic method that can be broadly implemented in Senegal, for example, based on RFLP-PCR and following a process that was successfully developed and implemented for the Ghanaian founder variant, *GJB2*: c.427C>T: p.(Arg143Trp), and included in the public-health-policy decisions in Ghana [39,40].

Contrary to data from Ghana, this study reported a high proportion of consanguinity (75.47%) that favored the enrichment of pathogenic variants, particularly *GJB2*: c.94C>T: p.(Arg32Cys), which accounted for 25% (*n* = 11/44 families). Like many west African countries, Senegal has several ethnic groups with a long tradition of consanguineous marriages. In two neighboring countries of Senegal, Mali and Mauritania, consanguinity accounted for 55.5% and 61.33% of familial cases of HI, respectively [23,35]. Consanguinity favors gene identification for numerous recessive conditions [41]. Given the high frequency of the variant among Senegalese consanguineous multiplex families, we postulate that c.94C>T: p.(Arg32Cys) may be a founder variant in Senegal. Future studies should explore this possibility. Indeed, recent data reported that *GJB2*: c.427C>T: p.(Arg143Trp) evolved in a single individual in Ghana about 10,000 years ago [42].

An unexpected finding was that the “Ghanaian” founder variant, i.e., *GJB2*: c.427C>T: p.(Arg143Trp), was present in 4.5% (*n* = 2/44 families) of multiplex families in Senegal. Interestingly, Ghana and Senegal do not share a border, and that variant in *GJB2* is absent in populations with HI from Nigeria [13], which is closer to Ghana. It is thus highly speculated that this finding is not due to regional migration, but rather to forced movement of people during the transatlantic slave trade. Indeed, slaves were brought to Gorée [43], an Atlantic island near the Senegalese coastal city of Dakar, before being transported to the Americas. Interestingly, the four families that segregated that variant were all based in Dakar. Future haplotype studies should comparatively investigate haplotypes in *GJB2*: c.427C>T: p.(Arg143Trp) in families from both Ghana and Senegal to explore this hypothesis. Moreover, the Mayan founder variant, *GJB2*: c.132G>A: p.(Trp44Ter), reported by Adadey SM et al. [12] in a Ghanaian family was also identified in a Senegalese family, in the compound heterozygous state.

There has been growing evidence of the association between the type of variant and the severity of HI. The degree of *GJB2*-associated HI depends on the degree of damage to the coding protein Cx26 [44]. Truncating variants, which create a premature stop codon and may result in the absence of any functional Cx26 protein, have been reported to induce a profound HI [45]. In our cohort, patients carrying *GJB2*: c.132G>A: p.(Trp44Ter), which is a non-sense variant, in a compound heterozygous state, exhibited a profound HI, which is in line with previous reporting in Guatemala [46]. However, the most common variant, *GJB2*: c.94C>T: p.(Arg32Cys) was associated with variable degrees of HI, ranging from moderate to profound. This variability may reflect a possible effect of modifier genes and/or environmental factors that lead to variable expression [47].

*GJB6* is located 50kb upstream of *GJB2*, and the del(*GJB6*-D13S1830) variant is the most common deletion of *GJB6* and is the second most prevalent ARNSHI variant in western European populations [17]. The deletion occurs in trans in either the homozygous or heterozygous state with pathogenic *GJB2* variants, and appears to have an ethnic-specific origin. The del(*GJB6*-D13S1830) variant was not found in our cohort of HI participants. This is in line with data reported from other African populations [12,16]. However, in a multicentric study, it has been shown that the *GJB6*-D13S1830 deletion is most frequent in Spain, France, the United Kingdom, Israel, and Brazil (5.9–9.7% of all DFNB1 alleles), is less frequent in the USA, Belgium, and Australia (1.3–4.5% of all DFNB1 alleles), and is very rare in southern Italy [10].

The study also indicates almost 2/3 of multiplex families with HI and all sporadic cases are eligible for next-generation sequencing, due to the highly heterogeneous genetic nature of NSHI. Future research should use high-throughput sequencing platforms that will allow the identification of pathogenic variants in either known genes or novel causative genes.

## 5. Conclusions

This is the first report of a genetic investigation of HI in Senegal. The study reveals a high consanguinity rate (75.47%) in affected families, and highlights that Senegal is the second country in SSA where *GJB2* pathogenic variants significantly contribute to ARNSHI, accounting for 15/44 (34.1%) in multiplex families. The data suggests that *GJB2*: c.94C>T: p.(Arg32Cys) and *GJB2*: c.427C>T: p.(Arg143Trp) should be tested in clinical practice for congenital HI in Senegal. Further studies using whole exome or whole genome sequencing approaches are needed to identify the other genes involved in families that are *GJB2* negative in Senegal.

## Figures and Tables

**Figure 1 biology-11-00795-f001:**
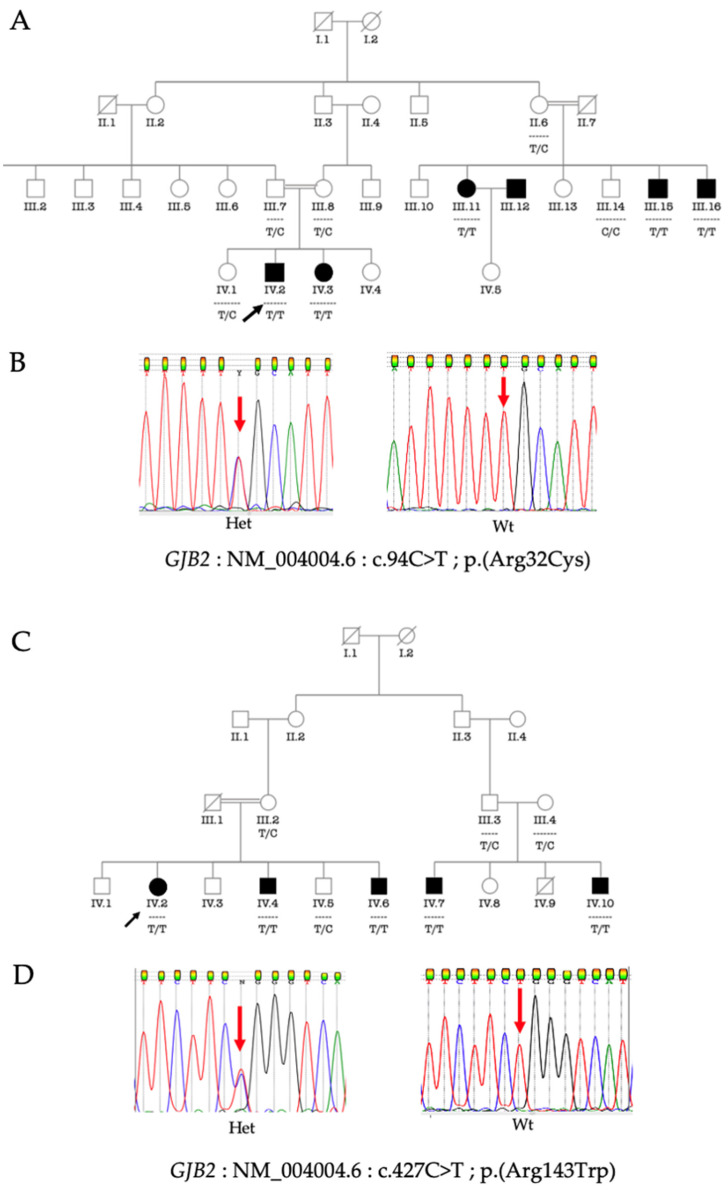
Pedigree of two multiplex families segregating HI and bi-allelic *GJB2*: c.94C>T: p.(Arg32Cys) and *GJB2*: c.427C>T: p.(Arg143Trp), respectively; black arrow indicates the proband (**A**,**C**). Electropherograms showing the reference and the pathogenic allele (**B**,**D**). The red arrows indicate the nucleotides affected by the variant. Het, heterozygous for the variant allele; Wt, wild type (homozygous for the reference allele) (**B**,**D**).

**Figure 2 biology-11-00795-f002:**
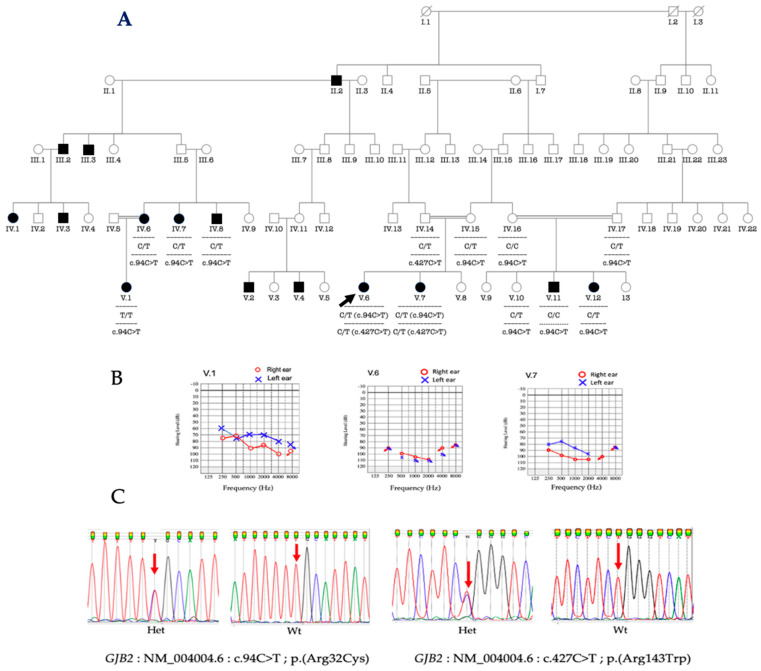
Pedigree of a multiplex family segregating HI with observed genotypes. V.6 is the proband (**A**); audiological phenotypes of the proband V.6, and her sister V.7, and the cousin from the father’s side, V.1 (**B**); electropherograms of pathogenic variants in *GJB2* (**C**); Het, heterozygous; Wt, wild type. Black arrow indicates the proband. The red arrows indicate the nucleotides affected by the variant; black arrow indicates the proband.

**Table 1 biology-11-00795-t001:** Repartition of patients according to the degree of HI and the mean age at medical diagnosis.

Degree of HI	Number of Patients(*n*)	Mean Age at Medical Diagnosis
Moderate (41–60 dB)	8 (6.42%)	8.37 ± 3.81 [5–14 years]
Severe (61–80 dB)	14 (11.29%)	4.25 ± 3.77 [1.5–13 years]
Profound (≥81 dB)	107 (82.26%)	2.33 ± 1.14 [1–6 years]

**Table 2 biology-11-00795-t002:** *GJB2* pathogenic variants among 15/44 multiplex families with congenital ARNSHI.

Genotypes	Multiplex Families	
	*n **	% (*n*/N)
[c.94C>T]; [c.94C>T]	11	25
[c.427C>T]; [c.427C>T]	2	4.54
[c.427C>T]; [c.94C>T]	1	2.27
[c.427C>T]; [c.132G>A]	1	2.27
Total	15	34.09

* Number of multiplex families.

**Table 3 biology-11-00795-t003:** Comparison of *GJB2* variants identified in Senegal and other populations from *Ensembl* database.

			Allele Frequency (*n/N*)			Allele Frequency from *Ensembl*				
Variants	rs Number	Allele	Cases	Controls	*p*-Value (Cases vs. Controls)	Global	Africa	America	East Asia	Europe
c.94C>T	rs371024165	C	0.78 (86/106)	0.99 (294/296)	<0.0001	1.0000	1.0000	1.0000	1.0000	1.0000
		T	0.22 (23/106)	0.01 (2/296)		0.0000	0.0000	0.0000	0.0000	0.0000
c.427C>T	rs80338948	C	0.94 (100/106)	0.98 (292/296)	0.024	0.9998	1.0000	1.0000	0.9990	1.0000
		T	0.06 (6/106)	0.02 (4/296)		0.0002	0.0000	0.0000	0.0010	0.0000
c.132G>A	rs104894407	G	0.99 (105/106)	0.996 (295/296)	0.458	0.9998	1.0000	1.0000	1.0000	1.0000
		A	0.01 (1/106)	0.004 (1/296)		0.0002	0.0000	0.0000	0.0000	0.0000

**Table 4 biology-11-00795-t004:** Comparison of *GJB2* genotypes and the degree of HI.

Genotypes	Degree of HI		
	Moderate (41–60 dB)	Severe (61–80 dB)	Profound (≥81 dB)
[c.94C>T]; [c.94C>T]	4	3	9
[c.427C>T]; [c.427C>T]	0	0	7
[c.427C>T]; [c.94C>T]	0	0	2
[c.427C>T]; [c.132G>A]	0	0	3

## Data Availability

The raw data supporting the conclusions of this article will be made available by the authors, without undue reservation.

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
