# Peer review of "GJB2 Is a Major Cause of Non-Syndromic Hearing Impairment in Senegal"

_biology, 2022, doi:10.3390/biology11050795_

Round 1

Reviewer 1 Report

The study is to investigate the association between GJB2/GJB6 variants and familial non-syndromic hearing impairment (HI) in Senegal, because the genetic etiology of ARNSHI in Senegal has not yet been investigated.  The authors investigated 129 affected and 143 unaffected individuals from 44 multiplex families segregating autosomal recessive non-syndromic HI, 9 sporadic HI cases of putative genetic origin, and 148 control individuals without personal or family history of HI and found high consanguinity rate in affected families.

I read this paper with great interest. To me, the paper is well written and organized.  In addition, the findings are significant to the clinical practice.   But I still have few comments and hope can improve the paper.

  1. Page 1, the Simple Summary looks redundant. I suggest that authors combine this section with Abstract and remove the duplicate information.
  2. To investigate the hypothesis, authors collected data from families in Senegal. I would like to see some discussion in the paper regarding how the data were sampled and how to alleviate the bias.
  3. Line 154, ‘Khi Square test’. I assume authors are talking about ‘Chi-square test’.  Please consider changing to 'Chi-square test' that is commonly used.
  4. Table 3. Firstly, please format the table with same font size and alignment etc. Secondly, are these p-values from ‘Chi-square test’? If so, since some observed numbers in couple of cells are less than 5, which will invalidate the test.  Please consider using the ‘Fisher’s exact test’ as alternative.  Thirdly, when report p-value less than 0.0001, ‘<.0001’ could be used.

Thank you.

Author Response

Comment 1: Page 1, the Simple Summary looks redundant. I suggest that authors combine this section with Abstract and remove the duplicate information.

Response:  Thank you for pointing this out. The inclusion of a simple Summary ins requirement of the journal, Biology. Redundancy is therefore expected.

Comment 2: To investigate the hypothesis, authors collected data from families in Senegal. I would like to see some discussion in the paper regarding how the data were sampled and how to alleviate the bias.

Response: Thanks for your comment. The recruitment of cases was based on families segregating HI in at least two affected individuals, and families were recruited nationwide in schools for the deaf, and within the communities, following a similar successful methods we previously implemented in both Cameroon, and Mali [16,23]. Therefore, we do not expect any important bias in the sampling of cases.  Nevertheless, the recruitment of apparently healthy controls, from blood donors, did not matched the geographical area where families were recruited.  Therefore the ethnolinguistic and geographical origin of control are likely not be representative of the general Senegalese population, and have probably biased the carrier frequency estimates for GJB2 - 427C>T; p.(Arg143Trp), and c.94C>T ; p.(Arg32Cys); this limitation should be alleviated in future studies.

Comment 3: Line 154, ‘Khi Square test’. I assume authors are talking about ‘Chi-square test’.  Please consider changing to 'Chi-square test' that is commonly used.

Response:  We have changed accordingly. Thanks!

Comment 4: Table 3. Firstly, please format the table with same font size and alignment etc. Secondly, are these p-values from ‘Chi-square test’? If so, since some observed numbers in couple of cells are less than 5, which will invalidate the test.  Please consider using the ‘Fisher’s exact test’ as alternative.  Thirdly, when report p-value less than 0.0001, ‘<.0001’ could be used.

Response 1: Thanks for the important comments. We have revised the method (line 154), and Table 3, accordingly.

Reviewer 2 Report

In this manuscript, the authors have elaborated an epidemiological study of deafness cases focusing on the GJB2 gene in Senegal population. Using sanger sequencing they have identified 4 mutations causing hearing loss. one of them, c.94C>T has been the most frequent in the studied population (25%). The Ghanaian mutation p.(Arg143Trp) was also identified.

The paper is well performed and presented and only several considerations should be taken into account:

1-Why have you not analyzed the possible presence of other deletions in the gene that have been previously described in the literature? these deletions could explain some cases of patients with heterozygous mutations.

2 the NM reference of the GJB2 gene should be indicated.

Author Response

Comments from Reviewer 2

Comment 1: Why have you not analyzed the possible presence of other deletions in the gene that have been previously described in the literature? these deletions could explain some cases of patients with heterozygous mutations.

Response: Thank you for this suggestion. We focused on GJB6-D13S1830 deletion since it is the most prevalent. The other deletions of GJB6 (D13S1834 and D13S1854) have not been reported in an African population yet. Moreover, we performed Whole Exome Sequencing for all the study participants (affected and unaffected) and no GJB6 deletion was identified.

Comment 2: the NM reference of the GJB2 gene should be indicated.

Response: Thank you for pointing this out. We have now added the GJB2 reference sequence (NM_004004.6.; retrieved from NCBI browser), as the MIM numbers, GJB2 (MIM: 121011) and GJB6 (MIM; 604418), when first mentioned in the text.

Round 2

Reviewer 1 Report

The revised version looks good.  No more comments.  Thanks!